# Unsupervised Distillation of Syntactic Information from Contextualized Word Representations

## Abstract

Contextualized word representations, such as ELMo and BERT, were shown to perform well on various semantic and structural (syntactic) task. In this work, we tackle the task of unsupervised disentanglement between semantics and structure in neural language representations: we aim to learn a transformation of the contextualized vectors, that discards the lexical semantics, but keeps the structural information. To this end, we automatically generate groups of sentences which are structurally similar but semantically different, and use metric-learning approach to learn a transformation that emphasizes the structural component that is encoded in the vectors. We demonstrate that our transformation clusters vectors in space by structural properties, rather than by lexical semantics. Finally, we demonstrate the utility of our distilled representations by showing that they outperform the original contextualized representations in a few-shot parsing setting.

## 1 Introduction

Human language[1] is a complex system, involving an intricate interplay between meaning (semantics) and structural rules between words and phrases (syntax). Self-supervised neural sequence models for text trained with a language modeling objective, such as ELMo (Peters et al., 2018), BERT (Devlin et al., 2019), and RoBERTA (Liu et al., 2019b), were shown to produce representations that excel in recovering both structure-related information (Gulordava et al., 2018; van Schijndel & Linzen; Wilcox et al., 2018; Goldberg, 2019) as well as in semantic information (Yang et al., 2019; Joshi et al., 2019).

In this work, we study the problem of disentangling structure from semantics in neural language representations: we aim to extract representations that capture the structural function of words and sentences, but which are not sensitive to their content. For example, consider the sentences:

1. Neural networks are interesting.    3. I study neural networks.
2. Maple syrup is delicious.    4. John loves maple syrup.

While (1) and (2) are different in content, they share a similar structure, the corresponding words in them, while unrelated in meaning,[2] serve the same function. Similarly for sentences (3) and (4). In contrast, sentence (1) shares the phrase *neural networks* with sentence (3) and *maple syrup* is shared between (2) and (4).[3] While the two occurrences of each phrase share the meaning, they are used in different structural (syntactic) configurations, serving different roles within the sentence (appearing in subject vs object position).[4] We seek a representation that will expose the similarity between "networks" in (1) and "syrup" in (2) while ignoring the similarity between "syrup" in (2) and "syrup" in (4).

---

[1] In this work we focus on English.

[2] We focus on lexical semantics.

[3] There is a syntactic distinction between the two, with "maple" being part of a noun compound and "neural" being an adjective. However, we focus in their similarity as noun modifiers in both phrases.

[4] These differences in syntactic position are also of relevance to language modeling, as different positions may pose different restrictions on the words that can appear in them.

We aim to learn a function from contextualized word representations to a space that exposes these similarities. Crucially, we aim to do this in an unsupervised manner: we do not want to inform the process of the kind of structural information we want to obtain. We do this by learning a transformation that attempts to remove the lexical-semantic information in a sentence, while trying to preserve structural properties.

Disentangling syntax from lexical semantics in word representations is a desired property for several reasons. From a purely scientific perspective, once disentanglement is achieved, one can better control for confounding factors and analyze the knowledge the model acquires, e.g. attributing the predictions of the model to one factor of variation while controlling for the other. In addition to explaining model predictions, such disentanglement can be useful for the comparison of the representations the model acquires to linguistic knowledge. From a more practical perspective, disentanglement can be a first step toward controlled generation/paraphrasing that considers only aspects of the structure, akin to the style-transfer works in computer vision, i.e., rewriting a sentence while preserving its structural properties while ignoring its meaning, or vice-versa. It can also inform search-based application in which one can search for "similar" texts while controlling various aspects of the desired similarity.

To achieve this goal, we begin with the intuition that the structural component in the representation (capturing the *form*) should remain the same regardless of the lexical semantics of the sentence (the *meaning*). Rather than beginning with a parsed corpus, we automatically generate a large number of structurally-similar sentences, without presupposing their formal structure (§3.1). This allows us to pose the disentanglement problem as a metric-learning problem: we aim to learn a transformation of the contextualized representation, which is *invariant* to changes in the lexical semantics within each group of structurally-similar sentences (§3.3). We demonstrate the structural properties captured by the resulting representations in several experiments (§4), among them automatic identification of structurally-similar words and few-shot parsing.

## 2 RELATED WORK

The problem of disentangling different sources of variation has long been studied in computer vision, and was recently applied to neural models (Bengio et al., 2013; Mathieu et al., 2016; Hadad et al., 2018). Such disentanglement can assist in learning representations that are invariant to specific factors, such as pose-invariant face-recognition (Peng et al., 2017) or style-invariant digit recognition (Narayanaswamy et al., 2017). From a generative point of view, disentanglement can be used to modify one aspect of the input (e.g., "style"), while keeping the other factors (e.g., "content") intact, as done in neural image style-transfer (Gatys, 2017).

In the field of NLP, disentanglement is much less researched. In controlled natural language generation and style transfer, several works attempted to disentangle factors of variation such as sentiment or age of the writer, with the intention to control for those factors and generate new sentences with specific properties, or transfer existing sentences to similar sentences that differ only in the those properties. Several works (Sohn et al., 2015; Ficler & Goldberg, 2017) have trained conditional generative models by explicitly conditioning a decoder network with a vector of attributes. On training, the attributes derive from the training sentence, while in testing the conditioning vector can be set to generate a text with the desired attributes. Other works Fu et al. (2018); Hu et al. (2017) aim to achieve style transfer (as opposed to generation) by explicitly training representations that are invariant to the controlled attributes (e.g. by co-training of a generator and attribute discriminator). A decoder generates the transferred text from the disentangled representation and from an explicit attribute representation. Lample et al. (2018) use a conditioned back-translation approach to achieve a similar goal. While these works try to disentangle sentence-level attributes, in this study we focus on disentangling between two components in the representations of individual words.

Several works examine the way semantic and syntactic information is distributed across the layers of neural models of text (Blevins et al., 2018; Tenney et al., 2019). They use diagnostic classifiers (Adi et al., 2016; Hupkes et al., 2018) to predict syntactic properties and demonstrate that different parts of the model encode information in different levels of abstraction (e.g. POS information, dependency label and semantic role). Liu et al. (2019a) used diagnostic classifiers trained to predict various syntactic and semantic properties from state-of-the-art LMs representations, and demonstrated that many syntactic and semantic distinctions are encoded in the probed representations. Clark et al.

(2019) probed the attention patterns of BERT, and showed that individual attention-heads focus in syntactically-meaningful relations in the input.

Beyond the descriptive level, recent works have focused on *supervised* extraction of syntax-related representations from neural representations. Artetxe et al. (2018) used a linear transformation, inspired from the notion of "similarity order" in classic word representation learning, to tailor *uncontextualized* word representations to syntactic vs. semantic tasks. Hewitt & Manning (2019) demonstrated that it is possible to train a linear transformation, under which squared euclidean distance between transformed contextualized word vectors correspond to the distances between the respective words in the syntax tree that represents the hierarchical structure of the sentence. Concurrent to this work, Li & Eisner (2019) have used a variational estimation method (Alemi et al., 2016) of the information-bottleneck principle (Tishby et al., 1999) to extract word embeddings that are useful to the end task of parsing.

While impressive, those works presuppose a specific formal syntactic structure (e.g. annotated parse trees following a specific linguistic annotation schema) and use this syntactic signal to learn structural information in a supervised manner. In other words, these works assume a given structure, and use supervision to make the structural information more salient, *mapping* (or *forcing*) the neural representations to known linguistic properties. In contrast, we aim to *expose* the structural information encoded in the network in an unsupervised manner, without pre-supposing an existing syntactic annotation scheme.

## 3 METHOD

Our goal is to learn a function $f : \mathbb{R}^n \mapsto \mathbb{R}^m$, which operates on contextualized word representations $x$ and extracts vectors $f(x)$ which make the structural information encoded in $x$ more salient, while discarding as much lexical information as possible. In the sentences "Maple syrup is delicious" and "Neural networks are interesting", we want to learn a $f$ such that $f(v^2_{\text{syrup}}) \approx f(v^1_{\text{networks}})$, where $v^i_{\text{word}}$ is the contextualized vector representation of the word in sentence $i$. We also want $f(v^4_{\text{syrup}}) \approx f(v^3_{\text{networks}})$, while keeping $f(v^1_{\text{networks}}) \not\approx f(v^3_{\text{networks}})$.

Moreover, we would like the *relation* between the words "maple" and "delicious" in the second sentence, to be similar to the relation between "neural" and "interesting" in the first sentence: $\text{pair}(v^2_{\text{maple}}, v^2_{\text{delicious}}) \approx \text{pair}(v^1_{\text{neural}}, v^1_{\text{interesting}})$. Operativly, we represent pairs of words $(x, y)$ by the difference between their transformation $f(x) - f(y)$, and aim to learn $f$ that preserves: $f(v^2_{\text{maple}}) - f(v^2_{\text{delicious}}) \approx f(v^1_{\text{neural}}) - f(v^1_{\text{interesting}})$. The choice to represent pairs this way was inspired by several works that demonstrated that nontrivial semantic and syntactic relations between uncontextualized word representations can be approximated by simple vector arithmetic (Mikolov et al., 2013a;b; Levy & Goldberg, 2014).

To learn $f$, we start with groups of sentences that the sentences within each group are known to share their structure but differ in their lexical semantics. We call the sentences in each group *structurally equivalent*. Figure 1 shows an example of two structurally equivalent sets. Acquiring such sets is challenging, especially if we do not assume a known syntactic formalism and cannot mine for sentences based on their observed tree structures. To this end, we automatically generate the sets starting with known sentences and sampling variants from a language model (§3.1). Our sentence-set generation procedure ensures that words from the same set that share an index also share their structural function. We call such words *corresponding*.

- When a train ticket is purchased, a contract is established
- When a travel document is acquired, a settlement is declared
- When a winning vehicle is obtained, a competition is introduced
- When a winning bid is announced, a winner is created

- Shapley participated in the ¨ great debate ¨ with heber d
- Khan joined in the ¨ silent discussion ¨ with e t
- Parker figured in the ¨ coming showdown ¨ with block leader
- Moore engaged in the ¨ modern struggle ¨ with joseph israel

Figure 1: Two groups of structurally-equivalent sentences. In each group, the first sentence is original sentence from Wikipedia, and the sentences below it were generated by the process of repeated BERT substitution. Some sets of corresponding words–that is, words that share the same structural function–are highlighted in the same color.

We now proceed to learn a function $f$ to map contextualized vectors of corresponding words (and the relations between them, as described above) to neighbouring points in the space.

We train $f$ such that the representation assigned to positive pairs — pairs that share indices and come from the same equivalent set — is distinguished from the representations of negative pairs — challenging pairs that come from different sentences, and thus do not share the structure of the original pair, but can, potentially, share their lexical meaning. We do so using Triplet loss, which pushes the representations of pairs coming from the same group closer together (§3.3). Figure 2 sketches the network.

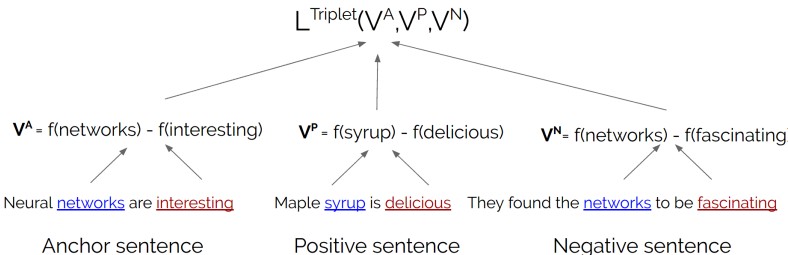

Figure 2: An illustration of triplet-loss calculation. Pairs of words are represented by the difference between their transformation $f$, which is identical for all words. The pairs of words in the anchor and positive sentences are lexically different, but structurally similar. The negative example presented here is especially challenging, as it is lexically similar, but structurally different.

### 3.1 GENERATING STRUCTURALLY-SIMILAR SENTENCES

In order to generate sentences that approximately share their structure, we sequentially replace content words in the sentence with other content words, while aiming to maintain the grammatically of the sentence, and keep its structure intact. Replacing words with words of the same POS, as done in Gulordava et al. (2018), does not answer to those requirements, as this method does not respect various restrictions that apply to words that share different function within the sentence. For example, verb argument structure dictates limitations on the arguments the predicate receives, and verbs differ in properties such as whether or not they accept a complement.

Replacing verbs with other verbs does not guarantee fulfilling these limitations. Since we do not want to rely on syntactic annotation (apart from the level of POS tags) when performing this replacement, we opted to use a pre-trained language model – BERT – under the assumption that strong neural language models do implicitly encode many of the syntactic restrictions that apply to words in different grammatical functions (e.g., we assume that BERT would not predict a transitive verb in the place of an intransitive verb, or a verb that accepts a complement in the place of a verb that does not accept a complement). While this assumption seems to hold with regard to basic distinctions such as transitive vs. intransitive verbs, its validity is less clear in the more nuanced cases, in which small differences in the surface level can translate to substantial differences in the deep structure – such as replacing a control verb with a raising verb. This is a limitation of the current approach, although we find that the average sentence we generate is grammatical and similar in structure to the original sentence. Moreover, as our goal is to *expose* the structural similarity encoded in neural language models, we find it reasonable to only capture the distinctions that are captured by a state-of-the-art neural language model.

**Implementation** Concretely, we rely on a BERT masked LM model. We start each group with a Wikipedia sentence, for which we generate $k = 6$ equivalent sentences by iterating over the sentence from left to right sequentially, masking the ith word, and replacing it with one of BERT's top-30 predictions. Crucially, to increase semantic variability, we perform the replacement in place (online), that is, after randomly choosing a guess $w$, we insert $w$ to the sentence at index $i$, and continue guessing the $i + 1$ word based on the modified sentence.[5] We exclude a closed set of a few dozens of words (mostly function words) and keep them unchanged in all $k$ variations of a

---

[5]We note that this process bears some similarity to Gibbs sampling from BERT conditioned LM.

sentence. To the extent that BERT learns to recover corrupted sentences by suggesting replacements that respect the probability distribution of actual natural language, the suggestions would be both semantically and structurally correct. We further maintain structural correctness by maintaining the POS, and encourage semantic diversity by the auto-regressive replacement process. We find that the average sentence is grammatical and maintain the structure of the original sentence, although some generated sentences violate these requirements. In the appendix §**??** we show some additional generated groups, and highlight some recurring errors.

The sets in Figure 1 were generated using this method.

### 3.2 WORD REPRESENTATION

We use the method to generate $N = 150,000$ equivalent sets $E_i$ of structurally equivalent sentences, and collect the contextualized vector representations of words in these sets, resulting in 1,500,000 training pairs and 200,000 evaluation pairs for the training process of $f$. We experiment with both ELMo and BERT-based contextualized representations. In average, we sample 11 pairs from each group of equivalent sentences. For ELMo, we represent each word in context as a concatenation of the last two ELMo layers (excluding the word embedding layer, which is not contextualized and therefore irrelevant for structure), resulting in representations of dimension 2048. For BERT, we concatenate the mean of the words' representation across all 23 layers of BERT-Large, with the representation of layer 16, which was found by Hewitt & Manning (2019) most indicative of syntax.

### 3.3 TRIPLET LOSS

We learn the mapping function $f$ using triplet loss (Figure 2).

Concretely, given a group of equivalent sentences $E_i$, we randomly choose two sentences to be the anchor sentence $S^A$, and the positive sentence $S^P$, and sample two different word indices $\{i_1, i_2\}$. Let $S^A[i_1]$ be the contextualized representation of the $i_1$th word in sentence $S^A$. The words $S^A[i_1]$ and $S^A[i_2]$ from the anchor sentence would form a representation of a pair of words, which should be close to the pair $S^P[i_1], S^P[i_2]$ from the positive sentence.

We represent pairs as their differences after transformation, resulting in the anchor pair $V^A$ and positive pair $V^P$:

$$V^A = f(S^A[i_1]) - f(S^A[i_2]) \qquad S^A \in E_i \tag{1}$$

$$V^P = f(S^P[i_1]) - f(S^P[i_2]) \qquad S^P \in E_i \tag{2}$$

where $f$ is the parameterized syntactic transformation we aim to learn. We also consider a negative pair:

$$V^N = f(S^N[j_1]) - f(S^N[j_2]) \qquad S^N \notin E_i \tag{3}$$

coming from sentence $S^N$ which is not in the equivalent set.

As $f$ has shared parameters for both words in the pair, it can thus be considered a part of a Siamese network, making our learning procedure an instance of a triplet Siamese network Schroff et al. (2015). We choose $f$ to be a simple model: a single linear layer that maps from dimensionality 2048 to 75. The dimensional of the transformation was chosen according to development set performance.

We use triplet loss (Schroff et al., 2015) to move the representation of the anchor vector $V^A$ closer to the representation of the positive vector $V^P$ and farther apart from the representation of the negative vector $V^N$. Following Hoffer & Ailon (2015), we calculate the softmax version of the triplet loss:

$$L^{triplet}(V^A, V^P, V^N) = \frac{e^{dist(V^A, V^P)}}{e^{dist(V^A, V^P)} + e^{dist(V^A, V^N)}} \tag{4}$$

where $dist(x, y) = 1 - \frac{x^\top y}{\|x\|\|y\|}$ is the cosine-distance between the vectors $x$ and $y$. Note that $L^{triplet} \to 0$ as $\frac{dist(V^A, V^P)}{dist(V^A, V^N)} \to 0$, as expected. The triplet objective is optimized end-to-end using

the Adam optimizer (Kingma & Ba, 2015). We train for 5 epochs with a mini-batch of size 500 [6], and take the last model as the final syntactic extractor. During training, the gradient backpropagates through the pair vectors to the parameters $f$ of the Siamese model, to get representations of individual words that are similar for corresponding words in equivalent sentences. We note that we do not back-propagate the gradient to the contextualized vectors: we keep them intact, and only adjust the learned transformation.

**Hard negative sampling** We obtain the negative vectors $V^N$ using hard negative sampling. For each mini-batch $B$, we collect 500 $\{V_i^A, V_i^P\}$ pairs, each pair taken from an equivalent set $E_i$. The negative instances $V_i^N$ are obtained by searching the batch for a vector that is closest to the anchor and comes from a different set:

$$V_i^N = \underset{V_{j \neq i}^A \in B}{\arg\min} \, dist(V_i^A, V_j^A) \tag{5}$$

where $dist$ is again the cosine distance. In addition, we enforce a symmetry between the anchor and positive vectors, by adding a pair (positive, anchor) for each pair (anchor, positive) in $B$.

That is, $V_i^N$ is the "most misleading" word-pair vector: it comes from a sentence that has a different structure than the structure of $V_i^A$ sentence, but is the closest to $V_i^A$ in the mini-batch [7].

## 4 EXPERIMENTS AND ANALYSIS

We have trained the syntactic transformation $f$ in a way that should encourage it to retain the structural information encoded in contextualized vectors, but discard other information. We assess the representations our model acquired in an unsupervised manner, by evaluating the extant to which the local neighbors of each transformed contextualized vector $f(x)$ share known structural properties, such as grammatical function within the sentence. For the baseline, we expect the neighbors of each vector to share a mix of semantic and syntactic properties. For the transformed vectors, we expect the neighbors to share mainly syntactic properties. Finally, we demonstrate that in a few-shot setting, our representations outperform the original ELMO representation, indicating they are indeed distilled from syntax, and discard other information that is encoded in ELMO vectors but is irrelevant for the extraction of the structure of a sentence.

**Corpus** For training the transformation $f$, we rely on 150,000 sentences from Wikipedia, tokenized and POS-tagged by spaCy [8]. The POS tags are used in the equivalent set generation to filter replacement words. Apart from POS tagging, we do not rely on any syntactic annotation during training. The evaluation sentences for the experiments mentioned below are sampled from a collection of 1,000,000 original and unmodified Wikipedia sentences (different from those used in the model training).

### 4.1 QUALITATIVE ANALYSIS

**t-SNE Visualization** Figure 3 shows a 2-dimensional t-SNE projection (Maaten & Hinton, 2008) of 15,000 random content words. The left panel projects the original ELMo states, while the right panel is the syntactically transformed ones. The points are colored according to the dependency label (relation to parent) of the corresponding word, assigned by the spacy parser.

As can be seen, in the original ELMo representation most states – apart from those characterized by a specific part-of-speech, such as amod (adjectives, in orange) or nummod (numbers, in light green) – do not fit well into a single cluster. In contrast, the syntactically transformed vectors are more neatly clustered, with some clusters, such as direct objects (brown) and prepositional-objects (blue), that are relatively separated after, but not before, the transformation. Interestingly, some functions that used to be a single group in ELMo (like the adjectives in orange, or the noun-compounds in

---

[6]A large enough mini-batch is necessary to find challenging negative examples.

[7]We implicitly assume that any pair coming from a different group of equivalent sentences is a valid negative example – that is, does not share the structural relation that exists between the anchor pair's words. This is a relatively mild assumption, as due to sparsity, in high probability two different sentences do not share the very same structure

[8]https://spacy.io/

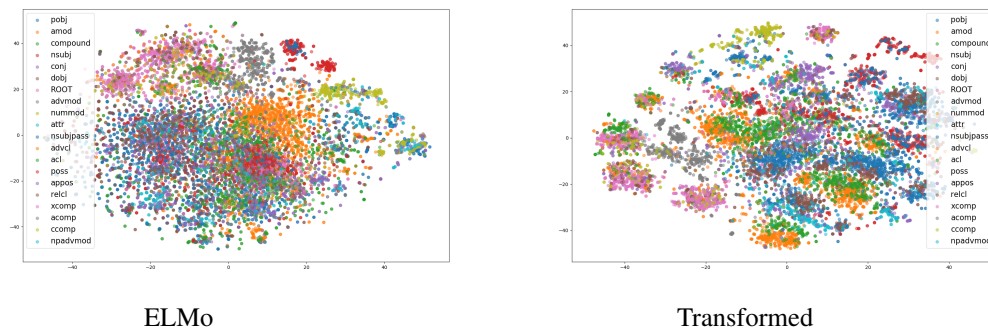

ELMo                                                    Transformed

Figure 3: t-SNE projection of ELMO states, colored by syntactic function, before (left) and after (right) the syntactic transformation.

green) are now split into several clusters, corresponding to their use in different sentence positions, separating for examples adjectives that are used in subject positions from those in object position or within prepositional phrases. Additionally, as noun compounds ("maple" in "maple syrup") and adjectival modifiers ("tasty" in "tasty syrup") are relatively structurally similar (they appear between determiners and nouns within noun phrases, and can move with the noun phrase to different positions), they are split and grouped together in the representation (the green and orange clouds).

To quantify the difference, we run $K$-means clustering on the projected vectors, and calculate the average cluster purity score as the relative proportion of the most common dependency label in each cluster. The higher this value is, the more the division to clusters reflect division to grammatical functions (dependency labels). We run the clustering with different $K$ values: 10, 20, 40, 80. We find an increase in class purity following our transformation: from scores of 22.6%, 26.8%, 32.6% and 36.4% (respectively) for the original vectors, to scores of 24.3%, 33.4%, 42.1% and 48.0% (respectively) for the transformed vectors.

**Examples**  Below are a few query words (Q) and their closest neighbours before (N) and after (NT) the transformation. Note the high structural similarity of the entire sentence, as well as the function of the word within it (Q1: last word of subject NP in a middle clause, Q2: possessed noun in sentence initial subject NP, Q3: head of relative clause of a direct object):

Q:*in this way of thinking, an impacting **projectile** goes into an ice-rich layer – but no further .*
N:*they generally have a pre-engraved rifling band to engage the rifled launch tube , spin-stabilizing the **projectile** , hence the term " rifle " .*
NT:*to achieve a large explosive yield, a linear implosion **weapon** needs more material, about 13 kgs.*

Q: *the mint 's **director** at the time , nicolas peinado , was also an architect and made the initial plans.*
N: *the **director** is angry at crazy loop and glares at him , even trying to get a woman to kick crazy loop out of the show ( which goes unsuccessfully ) .*
NT: *jetley 's **mother** , kaushaliya rani , was the daughter of high court advocate shivram jhingan .*

Q: *their first project is software that **lets** players connect the company 's controller to their device*
N: *you could try use norton safe web , which **lets** you enter a website and show whether there seems to be anything bad in it .*
NT: *the city offers a route-finding website that **allows** users to map personalized bike routes*

## 4.2 QUANTITATIVE EVALUATION

We expect our transformed vectors to capture more structural and less lexical similarities than the source vectors. We expect each vectors' neighbors in space to share the structural function of the word over which the vector was collected, but not necessarily share its lexical meaning. We focus on the following structural properties:

| | Dep. edge | Head's dep. edge | Tree path (complete) | Tree path (L=3) | Tree path (L=2) | Depth (correlation) | Lexical Match |
|---|---|---|---|---|---|---|---|
| Baseline (all) | 0.580 | 0.473 | 0.166 | 0.353 | 0.566 | 0.448 | 0.736 |
| Transformed (all) | 0.699 | 0.603 | 0.253 | 0.523 | 0.735 | 0.561 | 0.284 |
| Transformed-untrained (all) | 0.461 | 0.430 | 0.142 | 0.319 | 0.528 | 0.407 | 0.680 |
| Baseline (difficult) | 0.509 | 0.460 | 0.160 | 0.347 | 0.564 | 0.430 | 0.776 |
| Transformed (difficult) | 0.671 | 0.591 | 0.260 | 0.534 | 0.751 | 0.576 | 0.274 |

Table 1: Results in the closest-word queries, before and after the application of the syntactic transformation. "Basline" refers to unmodified ELMo vectors, "Transformed" refers to ELMo vectors after the learned syntactic transformation $f$, and "Transformed-untrained" refers to ElMo vectors, after a transformation that was trained on a randomely-initialized ELMo. "Difficult" refers to evaluation on the subset of POS tags which are most structurally diverse.

- Dependency-tree edge of a given word (dep-edge), that represents its function (subject, object etc.)

- The dependency edge of the word parent's (head's dep-edge) in the tree – to represent higher level structure, such as a subject that resides within a relative clause, as in the word 'man' in the phrase "the child that the man saw".

- Depth in the dependency tree (distance from the root of the sentence tree).

- Constituency-parse paths: Consider, for example, the sentence "They saw the moon with the telescope". The word "Telescope" is a part of a noun-phrase "The telescope", which resides inside a prepositional phrase "with the telescope", which is part of the Verbal phrase ""Saw with the telescope". The complete constituency path for this word is therefore "NP-PP-VP". We calculate the complete tree path to the root (Tree-path-complete), as well as paths limited to lengths 2 and 3.

For this evaluation, we parse 400,000 random sentences taken from the 1-million-sentences Wikipedia sample, run ELMo and BERT to collect the contextualized representations of the sentences' words, and randomly choose 400,000 query word vectors (excluding function words). We then retrieve, for each query vector $x$, the value vector $y$ that is closest to $x$ in cosine-distance, and record the percentage of closest-vector pairs $(x, y)$ that share each of the structural properties listed above. For the tree depth property, we calculate the Pearson correlation between the depths of the queries and the retrieved values. We use Spacy parser for dependency-parsing, and the Berkeley Neural Parser (Kitaev & Klein, 2018) for constituency parsing. We exclude function words from the evaluation.

**Easier and Harder cases**  The baseline models tend to retrieve words that are lexically similar. Since certain words tend to appear at above-chance probability in certain structural functions, this can make the baseline be "right for the wrong reason", as the success in the closest-word test reflects lexical similarity, rather than grammatical generalization of the model. To control for this confounding, we sort the different POS tags according to the entropy of their dependency-labels distribution, and repeat the evaluation only for words belonging to those POS tags having the highest entropy (those POS tags are the most structurally variant, and tend to appear in different structural functions). We find that the performance of the baselines (ELMo, BERT models) on those words drops significantly, while the performance of our model are only mildly influenced, further indicating the superiority of our model in capturing structural rather than lexical information.

**Results**  The results for ELMo are presented in Table 4. For BERT, we witnessed similar, but somewhat lower, accuracy: for example, 68.1% dependency-edge accuracy, 56.5% head's dependency-edge accuracy, and 22.1% complete constituency-path accuracy. The results for BERT are available in the appendix §C, and for the reminder of the paper, we focus in ELMo. We observe significant improvement over the baseline for all tests. The correlation between the depth in tree of the query and the value words, for examples, rises from 44.8% to 56.1%, indicating that our model encourages the structural property of the depth of the word to be more saliently encoded in its representation compared with the baseline. The most notable relative improvement is recorded with regard to full constituency-path to the root: from 16.6% before the structural transformation, to 25.3% after it – an improvement of 52%. In addition to the increase in syntax-related properties, we observe a sharp

drop – from 73.6% to 28.4% – – in the proportion of query-value pairs that are lexically identical (lexical match, Table 4). This indicates our transformation $f$ removes much of the lexical information, which is irrelevant for structure. To assess to what extent the improvements stems from the information encoded in ELMo, rather than being an artifact of the triplet-loss training, we also evaluate on a transformation $f$ that was trained on a randomly-initialized ELMo, a surprisingly strong baseline (Conneau et al., 2018). We find this model performs substantially worse than the baseline (Table 4, "Transformed-untrained (all)").

### 4.3 MINIMAL SUPERVISION FOR STRUCTURE DISTILLATION: FEW-SHOT PARSING

The absolute nearest-neighbour accuracy values may appear to be relatively low: for example, only 67.6% of the (query, value) pairs share the same dependency edge.

As the model acquires its representation without being exposed to human-mandated syntactic convention, some of the apparent discrepancies in nearest neighbours may be due to the fact the model acquires different kind of generalization, or learned a representation that emphasizes different kinds of similarities. Still, we expect the resulting (75 dimensional) representations to contain distilled structure information that is mappable to human notions of syntax. To test this, we compare dependency-parsers trained on our representation and on the source representation. If our representation indeed captures structural information, we expect it to excel on a low data regime. To this end, we test our hypothesis with few-shot dependency parsing setup, where we train a model to predict syntactic trees representation with only a few hundred labeled examples.

We use an off-the-shelf dependency parser (Dozat & Manning, 2016) and first swap the pre-trained Glove embeddings (Pennington et al., 2014) with ELMo contextualized embeddings (Peters et al., 2018). In order to have a fair comparison with our method, we use the concatenation of the two last layers of Elmo; we refer to this experiment as *elmo*. As our representation is much smaller than ELMo's (75 as opposed to 2048), a potential issue for a low data regime is the high parameter number to optimize in the later case, therefore a lower dimension can achieve better results. We design two additional baseline experiments to remedy this potential issue: (1) Using PCA in order to reduce the representation dimensionality. We randomly chose 1M words from Wikipedia, calculated their representation with ELMo embeddings and performed PCA. This transformation is applied during training on top of ELMo representation while keeping the 75 first components. This experiment is referred to as *elmo-pca*. This representation should perform well if the most salient information in the ELMo representations are structural. We exepct it to not be the case. (2) Automatically learning a matrix that reduces the embedding dimension. This matrix is learned during training and can potentially extract the relevant structural information from the representations. We refer to this experiment as *elmo-reduced*.

Lastly, we examine the performance of our representation, where we apply our structural extraction method on top of ELMo representation. We refer to this experiment as *syntax*.
We run the few-shot setup with multiple training size values: 50, 100, 200, 500. The results—for both labeled (LAS) and unlabeled (UAS) attachment scores—are presented in Figure 4, and the numerical results are available in the appendix §B.

We notice that in the lower training size regime, we obtain the best performances compared to all baselines. The more training data is used, the gap between our representation and the baselines reduced, but the *syntax* representation still outperforms *elmo*. Reducing the dimensions with PCA (*elmo-pca*) works considerably worse than ELMo, indicating that the most salient information is indeed not structural, and the PCA loses important information. Reducing the dimensions with a learned matrix (*elmo-reduced*) works substantially better than ELMo, and achieve the same UAS as our representation from 200 training sentences onward. However, our transformation was learned in an unsupervised fashion, without access to the syntactic trees. Finally, when considering the labeled attachment score, where the model is tasked at predicting not only the child-parent relation but also its label, our *syntax* representation outperforms *elmo-reduced*.

## 5 CONCLUSION

In this work, we propose an unsupervised method for the distillation of structural information from neural contextualized word representations. We used a process of sequential BERT-based substitu-

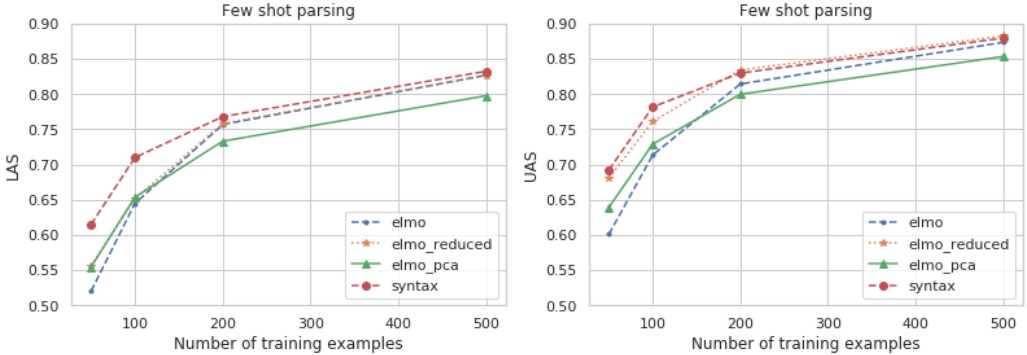

Figure 4: Results of the few shot parsing setup

tion to create a large number of sentences which are structurally similar, but semantically different. By controlling for one aspect – structure – while changing the other – lexical choice, we learn a metric (via triplet loss) under which pairs of words that come from structurally-similar sentences are close in space. We demonstrated that the representations acquired by this method share structural properties with their neighbors in space, and show that with a minimal supervision, those representations outperform ELMo in the task of few-shots parsing. The method presented here is a first step towards a better disentanglement between various kinds of information that is represented in neural sequence models.

The method used to create the structurally equivalent sentences can be useful by its own for other goals, such as augmenting parse-tree banks (which are often scarce and require large resources to annotate). In a future work, we aim to extend this method to allow for a more soft alignment between structurally-equivalent sentences.

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

## A   APPENDIX

## B   COMPLETE PARSING RESULTS

| Model | Number of sentences | | | |
|---|---|---|---|---|
| | 50 | 100 | 200 | 500 |
| ELMO | 0.52 | 0.64 | 0.75 | 0.82 |
| ELMO-reduced | 0.55 | 0.65 | 0.75 | 0.82 |
| ELMO-PCA | 0.55 | 0.65 | 0.73 | 0.79 |
| ELMO-syntax (ours) | 0.61 | 0.70 | 0.76 | 0.83 |

Table 2: Labeled parsing scores (LAS)

| Model | Number of sentences | | | |
|---|---|---|---|---|
| | 50 | 100 | 200 | 500 |
| ELMO | 0.60 | 0.71 | 0.81 | 0.87 |
| ELMO-reduced | 0.68 | 0.76 | 0.83 | 0.88 |
| ELMO-PCA | 0.63 | 0.72 | 0.79 | 0.85 |
| ELMO-syntax (ours) | 0.69 | 0.78 | 0.82 | 0.87 |

Table 3: Unlabeled parsing scores (UAS)

## C   BERT CLOSEST-WORD RESULTS

| | Dep. edge | Head's dep. edge | Tree path (complete) | Tree path (L=3) | Tree path (L=2) | Depth (correlation) | Lexical Match |
|---|---|---|---|---|---|---|---|
| Baseline (all) | 0.549 | 0.432 | 0.146 | 0.310 | 0.522 | 0.436 | 0.829 |
| Transformed (all) | 0.681 | 0.565 | 0.221 | 0.471 | 0.697 | 0.597 | 0.319 |
| Baseline (difficult) | 0.478 | 0.429 | 0.143 | 0.310 | 0.521 | 0.428 | 0.820 |
| Transformed (difficult) | 0.652 | 0.565 | 0.225 | 0.482 | 0.714 | 0.601 | 0.300 |

Table 4: Results in the closest-word queries, before and after the application of the syntactic transformation. "Basline" refers to unmodified vectors derived from BERT, and "Transformed" refers to the vectors after the learned syntactic transformation $f$. "Difficult" refers to evaluation on the subset of POS tags which are most structurally diverse.

