# OpenReview forum: "Unsupervised Distillation of Syntactic Information from Contextualized Word Representations"
_ICLR.cc/2020/Conference — Reject_

### Official Review · AnonReviewer2 · 2019-10-17
**Official Blind Review #2**

**Rating:** 6

**Review:**

Summary:
=========
This paper aims to disentangle semantics and syntax in contextualized word representations. The main idea is to learn a transformation of the contexualized representations that will make two word representations to be more similar if they appear in the same syntactic context, but less similar if they appear in different syntactic contexts. The efficacy of this transformation is evaluated through cluster analysis, showing that words better organize in syntactic clusters after the transformation, and through low-resource dependency parsing.

The paper presents a simple approach to transform word representations and expose their syntactic information. The experiments are mostly convincing. I would like to see better motivation, more engagement with a wider range of related work, and more thorough quantitative evaluations. Another important question to address is also what kind of semantic/syntactic types of information are targeted, and how to handle the tradeoff between them, for instance for different purposes.


Main comments:
==============
1. Motivation: I found the motivation for the problem understudied a bit lacking. The main motivation seems to be to disentangle semantic and syntactic information. But why should we care about that? Beyond reference to disentangling in computer vision, some more motivation would be good. The few-shot parsing is a good such motivation, although the results are a bit disappointing (see more on this below). Another possible motivation is potential applications of disentanglement in language generation. There is a line of work on style transfer also in language generation, and it seems plausible that the methodology could be applied to such tasks.
2. The present work is well-differentiated from work on extracting syntactic information from word representations via supervised ways, as the current work does so in an unsupervised way. I don't quite get the terminological differentiation between "mapping" and "extracting" in the introduction, but the idea is clear.
3. Have you considered alternative representations of word pairs besides the different of their transformations f(x)-f(y)?
4. I found it interesting that the word representation from BERT is the concatenation of layer 16 with the mean of all the other layers. This is motivated by Hewitt and Manning's findings, and [5] found similar results. However, the different between layer 16 and others is not that large as to warrant emphasizing it so much. Perhaps a scalar mix with fine-tuning may work better, as in [5], or another method. Have you tried other word representations? I also wonder whether it makes sense to use different layers for different parts of the triplet loss, depending on whether to emphasize syntactic vs. semantic similarity.
5. The introduction lays out connections to some related work, but leaves several relevant pieces missing. See examples below.
6. The results in 3.3 are limited but useful. The comparison with a PCA-ed and reduced representation is well thought of, because of the risk with low-resource and high dimensionality. That said, I found the gap between the proposed syntax model and the ELMo-reduced disappointingly small. Even in the LAS, it seems like the difference is very small, ~0.5, although it's hard to tell from the figure. Providing the actual numbers and a measure of statistical significance would be helpful here.
7. Some care should be taken to define what kind of semantics is targeted here. In several cases this is "lexical semantics", but then we have "meaning" in parentheses sometimes (end of intro). Obviously, there's much more to semantics and meaning that the lexical semantics, so a short discussion of how the work views other, say compositional semantics, would be good.


Other comments:
===============
1. The introduction seeks a representation that will ignore the similarity between "syrup" in (2) and (4). I wonder if "ignoring" is too strong. One may not want to lose all lexical semantic information. Moreover, the proposed triplet loss does not guarantee that information is ignored (and justly so, in my opinion).
2. In the example, "maple" and "neural" are said to be syntactically similar, although "maple syrup" is a noun compound while "neural networks" is an adjective-noun. Shouldn't they be treated differently then? Unless the notion of syntax is more narrow and just looks at unlabeled dependency arcs.
3. Some experimental choices are left unexplained, such as k=6 (section 2.1) or mapping to 27 dims (section 2.3); these two seem potentially important.
4. Section 2.3: do you also back-prop back into the BERT/ELMo model weights?
5. The dataset statistics in section 3 do not match those in section 2.2. Please clarify.
6. The qualitative cluster analysis via t-SNE (3.1) is compelling. It could be made stronger by reporting quantitative clustering statistics such as cluster purity before and after transformation.
7. In the examples showin in 3.1, it would be good to give also the nearest neighbor before the transformation for comparison.
8. The quantitative results in 3.2 convey the point convincingly. It's good to see also the lexical match measure going down. The random baseline is also a good sanity check to have. It would be good to provide full results with BERT, at least in the appendix and at least for section 3.2, maybe also for 3.3.
9. More related work:
+ Work that injects syntactic information into word representations in a supervised way, such as [1,2]
+ Work that shows that word embeddings contain different kinds of information (syntactic/semantic), and propose simle linear transformations to uncover them.
+ Engaging with the literature on style transfer in language generation would be good, as mentioned above for motivation, but also to situate this work w.r.t to related style transfer work.
+ Another line of work that may be mentioned is the variety of papers trying to extract syntactic information from contextualized word representations, such as constructing trees from attention weights. There were a few such papers in BlackboxNLP 2018 and 2019.

Typos, phrasing, formatting, etc.:
============================
- Abstract: a various of semantic... task -> various semantic... tasks; use metric-learning approach -> use a metric-learning approach; in few-shot parsing setting -> in a few-shot parsing setting
- Wilcox et al. does not have a year
- Introduction: few-shots parsing -> few-shot parsing
- Method: extract vectors -> extracts vectors; Operativly -> Operatively
- Section 3: should encourages -> should encourage; a few-shots settings -> a few-shot setting
- 3.2: -- was not rendered properly
- 3.3: matrix that reduce -> reduces


References
==========
[1] Levy and Goldberg. 2014. Dependency-Based Word Embeddings
[2] Bansal et al. 2014. Tailoring Continuous Word Representations for Dependency Parsing
[3] Artetxe et al. 2018. Uncovering divergent linguistic information in word embeddings with lessons for intrinsic and extrinsic evaluation
[4] Tenney et al. 2019. BERT Rediscovers the Classical NLP Pipeline
[5] Liu et al. 2019. Linguistic Knowledge and Transferability of Contextual Representations

**Experience Assessment:**

I have published in this field for several years.

**Review Assessment: Checking Correctness Of Derivations And Theory:**

I carefully checked the derivations and theory.

**Review Assessment: Checking Correctness Of Experiments:**

I carefully checked the experiments.

**Review Assessment: Thoroughness In Paper Reading:**

I read the paper thoroughly.

---

> ### Author Response · Authors · 2019-11-08
> **Response to "Other Comments"**
>
> # Responses to “Other comments”:
>
> 1. We agree that for many purposes is it beneficial not to discard the semantic information altogether. Ideally, we would want to allow “tuning” the representations on the semantics-syntax axis, changing the saliency of one factor or another according to the task at hand. However, as a first-order approximation, in this work we did aim to discard lexical semantics. The triplet objective does not explicitly do that, although insofar as the the “equivalent” sentences are indeed lexically diverse, we think that the hard-negatives mining should discard lexical semantics (as if vectors are embedded in space according to lexical information, the hard negatives would be semantically similar to the anchor vector, increasing the loss). But indeed, in practice we do not succeed in discarding all semantic information.
>
> 2. You are right, and we added a short discussion to the paper. We indeed observe that our method tends to conflate local syntactic distinctions like adjective-noun differences in environments where both can occur, in favor of more global distinctions (in this case, noun modifiers vs. other functions).
>
> 3. The dimensionality of the transformed vector was chosen according to development set performance. As for the value of K, it is not very important, as in practice we sample only 10 pairs from each group of sentences. This information was indeed missing in the paper. We added a clarification.
>
> 4. This is an important point, which we will make more salient in the text: we did not finetune the contextualized representations. We limited ourselves to extracting information that is already encoded in the vectors. This is consistent with our focus on disentangling existing representations rather than merely creating strong syntactic models.
>
> 5. The current description in the paper is indeed unclear. We will rephrase those parts. The number of evaluation sentences in section 2.2 refers to the training of the model with the triplet loss: we did not use them for the closest-word query. The 1 million sentences that are mentioned in section 3 are another (different) section of wikipedia, which we used for the closest-word test. From this set we sampled 400,000 sentences as mentioned in 3.2, and evaluated the percentage of (query, value) pair that share the different properties.
>
> 6. Thanks for the suggestion, we now added the purity measure to the section that presents the tsne results.
>
> 7. We added this comparison.
>
> 8. We added the closest-word results for BERT in an appendix.
>
> 9. Thanks for the references! We will definitely expand the discussion on related work. We will look further into the style transfer literature. If you are aware of references that are especially relevant, we would appreciate it if you share them.

---

> ### Author Response · Authors · 2019-11-08
> **Response to "Main Comments"**
>
> # Responses to “Main comments”:
>
> 1. Thank you for pointing this out, we strengthened the introduction on the motivation. As we see it, disentanglement is interesting for several reasons. From a purely scientific view, once disentanglement is achieved, one can better control for confounding factors and analyze the knowledge the model acquires, e.g.  attributing the predictions of the model to one factor of variation while controlling for the other. In addition to explaining model predictions, such disentanglement can be useful for the comparison of the representations the model acquires to linguistic knowledge, e.g. by showing the model is or is not able to learn specific linguistic abstractions, or by contrasting the way certain phenomena are represented in the model with their representation in syntactic schemes defined by linguists.The latter option can be especially illuminating, as the right way to describe certain syntactic phenomena is still in dispute among linguistics.
>
> From a more practical perspective, disentanglement can be a first step toward controlled generation/paraphrasing that considers only aspects of the structure, akin to the style-transfer works in computer vision. For example, one can imagine creating variants of a sentence that ignore syntax while preserving semantics, or that mimic the syntactic structures favoured by an authors. But we leave this to future research.
>
> 2. Thanks for pointing out to this, we tried to make the argument clearer this time.
>
> 3. We have considered alternatives, such as element-wise absolute value of the difference, element-wise multiplication, and an average. We did not observe qualitative differences between the different ways to represent pairs, and chose the difference because its simplicity and because of the known literature on arithmetic of word vectors, to which you have referred.
>
> 4. We experimented also with the last BERT layer alone, which performed worse. We added the full BERT results on the closest-word evaluation in the appendix. We agree that techniques such as a learned weighted average would probably yield better results and outperform ELMO. We did not try such methods due to time constraints. However, as we see the main contribution of the paper in the proposed method and in the demonstration of a proof-of-concept for unsupervised distillation of syntax, we think that the exact scores we get on, e.g., the parsing tasks, are relatively less important.
>
> 5. We expanded the existing background material into a separate related work section, in order to better situate this research with respect to previous works.
>
> 6. We added the exact numbers in the parsing experiments in an appendix. We agree that the differences are relatively small. However, with enough data, we cannot expect our representation to outperform ELMO, as our extraction process does not change the ELMO encoder but only transforms its output into a lower dimension, potentially discarding information. With enough direct supervision, the parser can probably directly extract the relevant information from ELMO vectors. We see the relatively significant LAS differences in very low data regime (50-100) as support for this hypothesis. We are not sure why the differences in the unlabeled setting are significantly smaller.
>
> 7. Thanks for pointing this out. We focus on lexical semantics, and we make it more explicit in the text.

---

> ### Comment · AnonReviewer2 · 2019-11-11
> **Thank you for your detailed response**
>
> I appreciate the detailed response to my review. I think the revised paper is improved in terms of background and motivation, and more complete experiments. It's also good to see the quantitative clustering purity results.
> While I don't think getting very high parsing results is a must for this work, I agree with reviewer 1 that comparing with a POS-based baseline is in order.
> I hope to see the paper accepted and at this point will keep my current evaluation.

---

> > ### Author Response · Authors · 2019-11-15
> > **Response**
> >
> > Thank you for the meaningful comments!

---

### Official Review · AnonReviewer1 · 2019-10-23
**Official Blind Review #1**

**Rating:** 8

**Review:**

The authors state a clear hypothesis: it is possible to extract syntactic information from contextualized word vectors in an unsupervised manner. The method of creating syntactically equivalent (but semantically different) sentences is indeed interesting on its own. Experiments do support the main hypothesis -- the distilled embeddings are stronger in syntactic tasks than the default contextualized vectors. The authors provide the code for ease of reproducibility which is nice.

There is a short literature review, but I am wondering if something similar was done for static word embeddings. I understand that they are obsolete these days, but on the other hand, they are better researched, so were there any attempts to disentangle syntax and semantics in the classical static word vectors?

Overall, I have no major concerns with the paper.

**Experience Assessment:**

I have published in this field for several years.

**Review Assessment: Checking Correctness Of Derivations And Theory:**

I carefully checked the derivations and theory.

**Review Assessment: Checking Correctness Of Experiments:**

I assessed the sensibility of the experiments.

**Review Assessment: Thoroughness In Paper Reading:**

I read the paper at least twice and used my best judgement in assessing the paper.

---

> ### Author Response · Authors · 2019-11-08
> **Response**
>
> Thank you for you comments!
>
> We are only aware of the work [1] which demonstrated the existence of a semantics-syntax tradeoff in word vectors that capture different orders of similarity. They projected pre-trained word vectors to different orders by a parameter-free transformation which derives from the similarity matrix, and measured the performance on semantic and syntactic tasks.  We now mention this work in the revised paper. We would appreciate pointers to other works in this direction, if any of the reviewers are aware of them.
>
> There are additional works which learn from scratch word embeddings that are tailored for syntax, some of them are mentioned by reviewer 2. These works are somewhat less relevant for the current study, as we aim to extract existing information from contextualized representations and make it more salient, rather than learning from scratch representations that capture syntax. Yet, we now note them in the new related work section.
>
> [1] Artetxe, Mikel, et al. "Uncovering divergent linguistic information in word embeddings with lessons for intrinsic and extrinsic evaluation." arXiv preprint arXiv:1809.02094 (2018). APA

---

### Official Review · AnonReviewer3 · 2019-10-26
**Official Blind Review #3**

**Rating:** 6

**Review:**

CONTRIBUTIONS:
Topic: Disentangling syntax from semantics in contextualized word representations
C1. A method for generating ‘structurally equivalent’ sentences is proposed, based only on the assumption that maintaining function words, and replacing one content word of a source sentence with another to produce a new grammatical sentence, yields a target sentence that is equivalent to the source sentence.
C2. The ‘structural relation’ between two words in a sentence is modeled as the difference between their vector embeddings.
C3a. The structural relation between a pair of content words in one sentence is assumed to be the same as that between the corresponding pair in an equivalent sentence.
C3b. The structural relation between any pair of content words in one sentence is assumed to be different from the structural relation between any pair of content words in an inequivalent sentence.
C4. Given a selected word in a source sentence, to generate an alternative ‘corresponding’ content word for an equivalent target sentence, BERT is used to predict the source word when it is masked, given the remaining words in the source sentence. The alternative corresponding word is randomly selected from among the top (30) candidates predicted by BERT. Given a source sentence, the set of target sentences formed by cumulatively replacing content words one at a time in randomly selected positions defines an ‘equivalence set’ in which words in different sentences with the same left-to-right index are corresponding words. (To promote the formation of grammatical target sentences, a word is only replaced by another word with the same POS.) A pre-defined set of equivalence sets is used for training.
C5. A metric learning paradigm with triplet loss is used to find a function f for mapping ELMo or BERT word embeddings to a new vector space of ‘transformed word representations’. Implementing C2 and C3a, given the indices i and i’ of two content words, the triplet loss rewards closeness of the difference D between the transformed embeddings of the pair of words with these indices in sentence S and the corresponding difference D’ for an equivalent sentence S’. Implementing C3b, the triplet loss penalizes closeness between D and D”, where D” is the difference between transformed word embeddings of a pair of content words in a sentence S” that is inequivalent to S. (Eq. 4).
C6. (Implementing C5.) To form a mini-batch for minimizing the triplet loss, a set of (500) sentences S is selected, and for each a pair of indices of content words is chosen. Training will use the difference in the transformed embeddings of the words in S with these indices: call this D, and call the set of these (500) D vectors B. For each sentence S in B, a ‘positive pair’ (D, D’) is generated, where D’ is the corresponding difference for S’, a selected sentence in the equivalence set of S. Closeness of D and D’ is rewarded by the triplet loss, implementing C3a. To implement C3b, a ‘negative pair’ (D, D”), for which closeness is penalized by the loss, is formed as follows. D” is the closest vector in B to D that is derived from a sentence S” that is not equivalent to S.
C7. 2-D t-SNE plots (seem to) show that relative to the original ELMo embeddings, the transformed embeddings cluster better by POS (Fig. 3). (No quantitative measure of this is provided, and the two plots are not easy to distinguish.)
C8. Pairs of closest ELMo vectors share syntactic (dependency parse) properties to a greater degree after transformation than before (Table 1). To check that this goes beyond merely POS-based closeness, the syntactic relations that least determine POS are examined separately, and the result remains. Furthermore, the proportion of pairs of closest vectors that are embeddings of the same word (in different contexts) drops from 77.6% to 27.4%, showing that the transformation reduces the influence of lexical-semantic similarity. Similar results hold for BERT embeddings, but to a lesser degree, so the paper focusses on ELMo.
C9. Few-shot parsing. Two dependency parsers are trained, one on ELMo embeddings, the other on their transformations (under the proposed method). In the small-data regime (less than 200 training examples), the transformed embeddings yield higher parser performance, even when the encoding size of the ELMo embeddings is reduced (from 2048 to 75) to match that of the transformed embeddings by either PCA or a learned linear mapping. (Fig. 4)
RATING: Weak accept
REASONS FOR RATING (SUMMARY). Using deep learning to create an encoding of syntactic structure with minimal supervision is an important goal and the paper proposes a clever way of doing this. The only ‘supervision’ here comes from (i) the function/content-word distinction (C1 above): two grammatical sentences are structurally equivalent if [but not only if] one can be derived from the other by replacing one content word with another; and (ii) filtering candidate replacement words to match the POS of the replaced word. BERT’s ability to guess a masked word is put to good use in providing suitable content word substitutions. The experimental results are rather convincing.
REVIEW (beyond the summary above)
C1. This assumption is famously not deemed to be true in linguistics, where the structural difference between ‘control’ and ‘raising’ verbs is basic Ling 101 material: see https://en.wikipedia.org/wiki/Control_(linguistics)#Control_vs._raising. This particular structural contrast illustrates how verbs can differ in their argument structure, without there being function words to signal the difference. So substituting *verbs* in particular may be non-ideal for the purposes of this work. Even the third example given by the authors in Sec. 3.1 illustrates a related  point, where function words do signal the contrast:  while the meaning of ‘let’ and ‘allow’ may be very similar, their argument structures differ, so that replacing ‘lets’ with ‘allows’ in the first sentence, or the reverse in the second sentence, produces ungrammatical results:
*their first project is software that *allows* players connect the company ’s controller to their device
*the city offers a route-finding website that *lets* users to map personalized bike routes
Therefore, contrary to the paper, relative to linguistic syntactic structure, it is not a good result that ‘lets’ in the original version of the first sentence is the closest neighbor in transformed embedding space to ‘allows’ in the second. Rather, it is probably meaning, not structure, that makes ‘let’ and ‘allow’ similar.
It would improve the paper to make note of this general concern with C1 and to provide a response.
On another point, an important premise of the proposed method (C2 above) is that differences in vector space embeddings encode relations; this has been used by a number of previous authors since the famous Mikolov, Yih & Zweig NAACL2013, and that work should be cited and discussed.

**Experience Assessment:**

I have published in this field for several years.

**Review Assessment: Checking Correctness Of Derivations And Theory:**

N/A

**Review Assessment: Checking Correctness Of Experiments:**

I assessed the sensibility of the experiments.

**Review Assessment: Thoroughness In Paper Reading:**

I read the paper thoroughly.

---

> ### Author Response · Authors · 2019-11-08
> **Response**
>
> We appreciate your constructive and detailed review!
>
> Your are right in noting that subtle differences in the surface level can mask substantial differences in the deep argument structure of the sentence, and that verbs are particularly sensitive in this respect. This is a limitation of the current approach, which we now acknowledge in the paper. Thanks for pointing this out.  Indeed, in general we can expect the replacement process to yield a grammatical sentence with equivalent structures only to the extent that BERT implicitly encodes the grammatical restrictions that apply to the masked word (i.e., we can only capture raising vs control distinction to the extent BERT-like LM can captures them). While BERT is a powerful LM -- and that is the reason we used it rather than simple POS-based replacement -- it may at times violates some of those restrictions. As you point out, this reasoning behind the substitution process and the premises we made were not clearly stated in the paper, and made it clearer in the revisioned version. However, we note that the average sentences we generate seem grammatical, and do not diverge much from the structure of the original sentence; we therefore think this method does at least approximate our end goal of generating grammatical sentences of the same structure.
> Moreover, we remind that our method attempts to uncover the structural information that is encoded in the neural LMs. Thus, we find it reasonable to not capture structural distinctions that are not reflected in current state-of-the-art neural LMs.
>
> Thank you for pointing out to the works on vector-space arithmetic. This was our motivation for representing pairs as the difference between the corresponding word vectors, and we will explicitly mention that in the paper.

---

> > ### Comment · AnonReviewer3 · 2019-11-15
> > **AnonReviewer3 Response**
> >
> > I am satisfied with the responses to my review and the others. I am raising my rating to 8: Accept.

---

> > > ### Author Response · Authors · 2019-11-15
> > > **Response**
> > >
> > > Thank you for your constructive review and appreciation of our work!

---

### Official Review · AnonReviewer4 · 2019-10-27
**Official Blind Review #4**

**Rating:** 1

**Review:**

Summary
The authors proposed to disentangle syntactic information and semantic information from pre-trained contextualized word representations.

They use BERT to generate groups of sentences that are structurally similar (have the same POS tag for each word) but semantically different. Then they use a metric-learning approach to learn a linear transformation that encourages sentences from the same group to have closer distance. Specifically, they defined a triplet loss (Eq4) and uses negative sampling.

They use 150,000 sentences from Wikipedia to train the transformation. POS tags are obtained from spaCy. To evaluate the learned representations, they provided a tSNE visualization of the original and transformed representations (groups by dependency label); evaluate whether the nearest neighbor shares the same syntactic role; low-resource parsing.

Reasons of rejection:
1. I don't agree with the authors' argument, "we aim to extract the structural information encoded in the network in an unsupervised manner, without pre-supposing an existing syntactic annotation scheme".  First, what do you mean by structural information without a clear definition? Also, in the method, the authors construct a dataset where each group of the sentence share similar syntactic structures (having the same POS tag). It seems there that the structural information just means POS tags.

2. The author failed to convince me that the learned representation is more powerful than just combining POS tags with the original representations. Since POS tags are assumed to be available during training. I think a reasonable baseline in all experiments would be the performance based on POS tags. For example, in Figure 3, although the original EMLo representation does not correlates with the dependency label very much, the POS tags may do. In Figure 4, the authors should compare with delexicalized dependency parsing, which performs pretty well in los-resource setting.


**Experience Assessment:**

I have published one or two papers in this area.

**Review Assessment: Checking Correctness Of Derivations And Theory:**

I carefully checked the derivations and theory.

**Review Assessment: Checking Correctness Of Experiments:**

I carefully checked the experiments.

**Review Assessment: Thoroughness In Paper Reading:**

I read the paper thoroughly.

---

> ### Author Response · Authors · 2019-11-08
> **Response**
>
> Thank you for your comments.
>
> >>in the method, the authors construct a dataset where each group of the sentence share similar syntactic structures (having the same POS tag). It seems there that the structural information just means POS tags.
>
> Our method definitely captures more than the POS tag of the words. For example, it clearly differentiates nouns in subject position from nouns in object position from nouns in relative clauses, and also differentiates nouns in passive constructions from those in active voice. This holds also for multiple usages of the same word, e.g the word "dog" will get different neighbours in "the dog barked" and "he heard the dog bark", even though both are nouns. Similarly for verbs, adjectives, and other classes.
>
> As for the use of POS tags in the process of sentence generation, we agree this injects some syntactic bias to the model. Following the submission, we have performed experiments on datasets that are constructed without this using POS information, and got similar results, suggesting that the POS information is not necessary for our method. We need to work more to get detailed and robust results we can report, but we are in the process of doing so and will include these results in the camera ready version.
>
> >>what do you mean by structural information without a clear definition?
>
> By “Structural information” we refer to properties that linguists would identify as “syntax”. We did not want to pre-specify the characteristics of this structural representation (e.g. by identifying it with a certain type of dependency or constituency representation scheme), as many competing frameworks exist, and the “right” ways to represent different phenomena are still in dispute among linguists. As discussed in the paper, defining the problem in an implicit manner (structure A is similar to B and C is similar to D, without specifying what constitutes this similarity) allows us to not rely on any specific annotation scheme, but rather extract the structural representations in an unsupervised way. However, in evaluation, we do compare our representations with linguistic notions of syntax, and show that to a large degree our representations capture those properties, although they were not trained to explicitly achieve this objective.
>
> >>In Figure 4, the authors should compare with delexicalized dependency parsing, which performs pretty well in los-resource setting.
>
> Thank you for the suggestion. In this experiment, we have tried to cautionaly use controls: performing PCA on the ELMO vectors, and projecting them to a lower dimension using a learned linear transformation. The rationale behind these controls is to encourage the parser to discard irrelevant information from the ELMO vectors. We agree that a comparison to a POS-based parser is of interest, and can help test the claim we capture information beyond the POS level. We plan to perform this experiment for the camera ready version.

---

> > ### Comment · AnonReviewer4 · 2019-11-15
> > **Reply to response**
> >
> > Thank you for responding. After a second *very close* reading of the updated paper and the authors' reply, I maintain that this paper is far from ICLR standard and will keep my score at 1.
> >
> > 1. The authors propose to distill syntactic knowledge from the contextualized representation. However, the authors do not formalize the notion of "syntactic knowledge," and it is unclear what exactly they hope to disentangle from these representations.
> > 2. The authors approached this task by generating "syntactically similar" (also vaguely defined) sentence pairs using heuristics on BERT representations (Section 3.1). Given that this process is heuristic-drive, one would expect an analysis of the hyperparameters selected to achieve the goal. However, the authors performed no such analysis, and the hyperparameters of this generation process (k=6, top-30, ...) appear to be selected at random.
> > 3. The authors fail to convince me that their method has any practical utility (Section 4.3):
> >     a. Lack of a standard baseline. The authors do not address my concerns regarding the delexicalized parser baseline in the updated paper.
> >     b. Unfair comparisons. In Figure 4, the author's proposed method "Syntax" uses BERT during training (for generating sentence pairs). It is unfair that their baselines only have access to ELMo embeddings. A standard fine-tuned BERT should be the minimum comparison.
> >     c. Lack of standard datasets + automatically generated golden labels. The authors did not perform experiments on standard parsing datasets, and they fail to describe their parsing corpus in detail. According to Section 4 [Corpus], they used off-the-shelf spaCy parser to generate golden labels for 1M Wikipedia sentences. This practice is non-existent in literature. If weak-supervision (spaCy's output) is taken as golden labels, the authors should at least provide detailed statistics on their data, as well human-evaluation of the quality of those labels.

---

> > > ### Author Response · Authors · 2019-11-15
> > > **Response to Reply to response**
> > >
> > > Thanks for taking the time to carefully re-read our paper. We regret that you still do not "get" what we were trying to achieve in this work. Most notably, we were *not* aiming at beating any other system. That is simply not the intention of the works. Our intention was to distill the structural representation encoded in contextualized vectors, in an unsupervised manner. That is, to produce a representation that captures as much as possible of the structure of the sentence and as little as possible of its lexical semantics.
> > >
> > > Our experiments are intended to measure how well this goal (to preserve structural properties) was achieved. As no other work that we are aware of tackled this goal before, the claims of "unfair comparison" (points 3a,b,c) seem irrelevant.
> > >
> > > Responses to specific comments:
> > > 2) as we are not trying to beat anyone, we also see no reason to explore hyper-parameters. We chose an initial setup (indeed, heuristically) that worked fine for our purposes, and left it at that.
> > > 3a) while we do not see it as a baseline, we did agree that the delexicalized setup is interesting. As we wrote in our response, we will perform these experiments for the final version of the paper (we didn't have time to properly do this during the response period, as the author responsible for the parsing experiments was travelling. But this will happen for camera ready).
> > > 3b) we are not trying to beat ELMo (or BERT), we are trying to figure out what is captured by them.
> > > 3c) The 1M wikipedia sentences are not weak supervision, they are used for evaluation: we perform clustering and search for nearest neighbours in over this space. While this setup is not standard in the literature, our goal is also non standard. We are not trying to "win" a parsing context, but to distill syntactic knowledge from a pre-trained LM.
> > > 3c.2) We did have 150,000 sentences that were POS-tagged by spaCy and used for deriving the training instances. As mentioned in our response to other reviewers, after submission we also experimented with a version that did not use the POS-tag information, getting similar results.

---

### Author Response · Authors · 2019-11-08
**General response.**

We thank all reviewers for their insightful comments. We updated the paper to account for some of them. Other comments will require more time to properly address, but we are working towards that as well.

We address individual reviewers comments in the responses to their reviews.

---

### Decision · Program_Chairs · 2019-12-19

**Decision:**

Reject

**Comment:**

This paper aims to disentangle semantics and syntax inside of popular contextualized word embedding models. They use the model to generate sentences which are structurally similar but semantically different.

This paper generated a lot of discussion. The reviewers do like the method for generating structurally similar sentences, and the triplet loss.  They felt the evaluation methods were clever.  However, one reviewer raised several issues.  First, they thought the idea of syntax had not been well defined. They also thought the evaluation did not support the claims.  The reviewer also argued very hard for the need to compare performance to SOTA models.  The authors argued that beating SOTA is not the goal of their work, rather it is to understand what SOTA models are doing.  The reviewers also argue that nearest neighbors is not a good method for evaluating the syntactic information in the representations.

I hope all of the comments of the reviewers will help improve the paper as it is revised for a future submission.